# Effect of Iodide-Based Organic Salts and Ionic Liquid Additives in Dye-Sensitized Solar Cell Performance

**DOI:** 10.3390/nano12172988

**Published:** 2022-08-29

**Authors:** João Sarrato, Ana Lucia Pinto, Hugo Cruz, Noémi Jordão, Gabriela Malta, Paula S. Branco, J. Carlos Lima, Luis C. Branco

**Affiliations:** LAQV-REQUIMTE, Department of Chemistry, NOVA School of Science and Technology, FCT NOVA, Universidade NOVA de Lisboa, 2829-516 Caparica, Portugal

**Keywords:** dye-sensitized solar cells, ionic liquids and organic salts, imidazolium, picolinium, guanidinium, tetralkylammonium

## Abstract

The use of ionic liquid and organic salts as additives for electrolyte systems in dye-sensitized solar cells have been widely described in recent years. The tunability of their physical-chemical properties according to the cation–anion selection contributes toward their high efficiencies. For this purpose, several iodide-based organic salts including imidazolium, picolinium, guanidinium and alkylammonium cations were tested using acetonitrile/valeronitrile electrolytes and their photovoltaic parameters were compared. A best efficiency of 4.48% (4.15% for the reference) was found for 1-ethyl-2,3-dimethylimidazolium iodide ([C2DMIM]I) containing electrolyte, reaffirming the effectiveness of these additives. 4-tertbutylpyridine was included into the formulation to further improve the performance while determining which iodide salts demonstrate the highest synergy with this additive. [C2DMIM]I once again proved to be the superior additive, achieving an efficiency of 6.48% (6% for the reference). Electrochemical impedance spectroscopy was employed to elucidate the effects of the various additives, demonstrating the relevance of the counter electrode resistance on device performance. Finally, several computational descriptors for the cationic structures were calculated and correlated with the photovoltaic and resistance parameters, showing that properties related to polarity, namely relative positive charge, molecular polarizability and partition coefficient are in good agreement with the counter-electrode resistance.

## 1. Introduction

Dye-Sensitized Solar Cells (DSSCs) have proven themselves as a viable and sustainable alternative to conventional photovoltaic technologies, particularly for indoor [1,2] and window integration applications [3], given their ease of processing, lower cost as well as high substrate and color flexibility.

These devices operate based on small-molecule photosensitizers adsorbed on the surface of a highly porous semiconducting material, typically TiO_2_ (Figure 1), a mediating electrolyte and a counter electrode. Upon light absorption (1), the excited valence electrons of the dye are quickly injected into the conduction band (CB) of the semiconductor (2) and transported to the anode (3). The electric current then crosses an external circuit (4) and arrives at the cathode, where a catalytically active material such as Pt reduces the redox pair acting as the charge-carrier present in the electrolyte (5). Finally, after diffusing across the electrolyte, the charge-carrier regenerates the photo-oxidized dye (6), closing the electrochemical cycle of the device.

Given this operating architecture, it comes as no surprise that the judicious choice of electrolyte constitutes one of the most important aspects in achieving high efficiency DSSCs. This can vary from the choice of redox mediator, such as cobalt (II/III) [4,5] or copper (I/II) [6,7] complexes in place of I^−^/I_3_^−^, the use of gel-based quasi-solid state electrolytes [8,9,10] or even hole transport materials [11,12]. Despite this, the highest DSSC efficiency reported to date is that of a device employing a liquid electrolyte [13], and as such the further optimization of this configuration is a pressing concern.

One of the key disadvantages of liquid electrolytes is related to solvent evaporation or leakage, which significantly reduces the efficiency of the DSSC device [14]. To circumvent this issue, several higher boiling point solvents such as 3-methoxyproprionitrile (MPN) [15], N-methyl-2-pirrolidone (NMP) [16], ethylene or propylene carbonate [17,18], as well as mixtures with the more typically used acetonitrile (ACN) have been tested [19,20].

One class of solvents that seems promissory as alternative electrolytes are ionic liquids (ILs), given their non-flammability, exceedingly low vapor-pressure and high chemical and thermal stability. In addition to these favorable characteristics, the wide array of available cationic and anionic structures allows for simple fine-tuning of solvent properties. ILs have in fact been widely employed in DSSCs [21,22,23,24] (Table 1, top), with one of the very first reported stable devices possessing 1-methyl-3-hexylimidazolium iodide as the electrolyte [25].

Despite the excellent long stability results, ILs generally demonstrate lower performance when compared with typical organic solvents. This fact is mostly justified with the much higher viscosities of pure ILs, leading to significantly lower diffusion coefficients of the redox mediators and, consequently, lower regeneration of the photo-oxidized dye [29]. One countermeasure for this mass transport limitation is the use of binary or even ternary mixtures of ILs. A successful example of this approach is a report by Bai et al. [22], using a low viscosity IL 1-ethyl-3-methylimidazolium tetracyanoborate ([EMIM][TCB]) as well as a eutectic mixture of 1,3-dimethylimidazolium iodide ([DMIM]I) and 1-methyl-3-propylimidazolium iodide ([C3MIM]I) to achieve an efficiency of 8.2%. By comparison, the same eutectic system employing 1-allyl-3-methylimidazolium iodide ([AMI]I) only reached 7.1%, further underlining the importance of electrolyte viscosity.

Regardless of this improvement, the most efficient devices still correspond to electrolytes containing optimized concentrations of IL as an additive or alternative iodide source (Table 1, bottom) [26,27,28]. One of the first and most successful instances of this type of electrolyte mixture was employed by Nazeeruddin et al. [30], where 0.6 M 1-butyl-3-methylimidazolium iodide ([C4MIM]I) in ACN/valeronitrile (VCN) (85/15 *v*/*v*%) was used to achieve a very impressive efficiency of 11.2%. In a separate work by Gao et al. [26], a direct comparison between electrolyte mixtures containing ILs as mixed solvents or as additives was described, where the former achieved an efficiency of 7.4% and the latter 11.0%. This difference was found by electrochemical impedance spectroscopy (EIS) to be a consequence of the much lower diffusion coefficient of the IL-based electrolyte when compared to the organic solvent mixture. Additionally, transient photoelectrochemical measurements indicate a higher recombination rate for the former, presumably because of higher triiodide concentration on the photoanode surface. Both phenomena can be attributed to the previously addressed higher viscosities, which despite being lower for the [EMIM] [TCB] containing mixtures than pure IL, still cannot compete with organic solvent systems.

Currently, the presence of IL additives, particularly [DMIM]I or [C3MIM]I, in electrolyte formulations is near ubiquitous for the most efficient I^−^/I_3_^−^ based systems [31,32,33], and even in small concentrations (0.01 M) in the Co(II/III)-based electrolyte of the highest efficiency reported to date [13]. Given their prevalence, it is important to ascertain which ILs result in the most significant improvements in conversion efficiency, and how their structure directly affects the photovoltaic parameters. As such, several iodide-based organic salts, including alkylimidazolium, picolinium, guanidinium and tetra-alkylammonium possessing various substituents have been prepared and employed as electrolyte additives in order to evaluate their comparative performance. In addition, the effect of 4-tertbutylpyridine (4-TBP) addition was also investigated and clarified by EIS, where the resulting parameters can be correlated with various structural descriptors.

## 2. Materials and Methods

### 2.1. General Information and Instruments

All of the employed solvents and reagents were obtained commercially and used without further purification. Drying of the solvents was achieved as described by Bradley et al. [34], by using with M2A molecular sieves. The ^1^H and ^13^C-NMR spectra were acquired at 400 and 101 MHz, respectively, with a Bruker Avance III 400. ESI-MS spectra were obtained with a MS Agilent 6130B Single Quadrupole mass spectrometer after liquid chromatography purification with an LC Agilent 1200 Series system, with H_2_O/MeOH eluent and a 0.4 mL/min flow.

### 2.2. DSSC Fabrication and Photovoltaic Characterization

The preparation procedure has been previously described [35]. The conductive FTO glass (TEC7, Greatcell Solar, Queanbeyan, Australia) used was cleaned with detergent water, followed by water and then ethanol. For the preparation of anodes, a blocking layer was first prepared by immersing washed plates with an area of 15 cm × 4 cm in a 40 mM TiCl_4_/water solution at 70 °C for 30 min, followed by washing with water and ethanol. These were then sintered at 500 °C for 30 min. This layer improves adherence of subsequent TiO_2_ layers and serves to prevent recombination of electrons in the FTO with the redox mediator. Afterwards, transparent TiO_2_ layers were deposited on the previously prepared FTO plates by screen-printing a commercially available titania paste (18NR-T, Greatcell Solar, Queanbeyan, Australia) by employing a 43.80 mesh per cm^2^ polyester fiber frame. After coating, the films were dried at 125 °C on a hot plate, and the coating/dried step was repeated one more time. The coated plates then underwent several gradual steps: 325 °C over the course of 20 min, then 375 °C in 5 min and finally 500 °C over 30 min, remaining at this temperature for 15 min, after which they were slowly cooled to room temperature. Once again, a TiCl_4_/water solution treatment was performed as mentioned above, to increase the roughness of the TiO_2_ surface for better dye adsorption. The final step in the anode preparation involved screen-printing deposition of a commercially available reflective TiO_2_ paste (WER2-O, Greatcell Solar, Queanbeyan, Australia), followed by the same step-wise heating/sintering step. The inclusion of this scattering layer helps further improve photocurrent by scattering photons and allowing their absorption. The large 15 cm × 4 cm plates were then cut into 2 cm × 1.5 cm pieces with a circular anodearea of 0.196 cm^2^ and a thickness of 15 μm. These individual anodes were then soaked in the dark at room temperature for 2 h in a 0.5 mM ethanolic solution of N719 dye. Excess dye was rinsed off the photoanodes with absolute ethanol.

The counter-electrodes consisted of FTO glass plates with an area of 2 cm × 2 cm in which a hole was drilled. These were then cleaned with water and ethanol to wash away any glass or contaminants. A transparent Pt layer (PT1, Greatcell Solar, Queanbeyan, Australia) was then deposited on the conductive side of the FTO plate by doctor blade: approximately 0.5 cm of the glass edge was covered with adhesive tape (3 M Magic) to control the thickness of the deposited film and to mask an electric contact strip. The paste was then spread on the substrate uniformly by sliding a glass rod along the tape spacer. After removing the strip of tape, the plates were gradually heated over the course of 30 min to a temperature of 550 °C, where they remained for 30 min. The cathode and the previously prepared photoanode were assembled vertically with a hot melt gasket made of Surlyn ionomer (Meltonix 1170–25, Solaronix SA, Aubonne, Switzerland) into a sandwich-type arrangement and sealed with a thermopress. For the preparation of the electrolyte, the redox couple (I^−^/I_3_^−^) as LiI (0.5 M), IL (0.5 M) and I_2_ (0.05 M) was dissolved in a can/VCN (85:15, % *v*/*v*) mixture. The inclusion of a percentage of VCN as opposed to the exclusive use of ACN reduces the volatility of the electrolyte. In the case of the cells employing 4-TBP, a concentration of 0.5 M was included in addition to the previous components. For the reference cells, the IL was omitted and 1 M LiI was used instead. The drill present in the cathode allowed the injection of the electrolyte into the cell via backfilling under vacuum. Finally, the hole was sealed with adhesive tape.

Two cells were assembled for each compound under identical conditions, with the efficiencies being measured 5 times for each cell for a total of 10 measurements per compound.

A digital Keithley SourceMeter multimeter (PVIV-1A) connected to a PC was used to record the current density-voltage curves under simulated sunlight irradiation from an Oriel solar simulator (Model LCS-100 Small Area Sol1A, 300 W Xe Arc lamp equipped with AM 1.5 filter, 100 mW/cm^2^). The thickness of the TiO_2_ photoanode films was measured with an Alpha-Step D600 Stylus Profiler (KLA-Tencor, Milpitas, CA, USA).

### 2.3. Electrochemical Impedance Spectroscopy

EIS was performed in the DSSC in the dark using an Autolab PGSTAT 12 potentiostat/galvanostat, controlled with GPES/FRA2 software version 4.9 (Eco-Chemie, B.V. Software, Utrecht, The Netherlands) by applying a controlled potential open circuit potential and other potentials with a step potential of 0.1 V and in a frequency range from 10^−2^ to 10^6^ Hz for the characterization of the material. The obtained response can be represented as Nyquist plots, where the imaginary part of the impedance is plotted vs. the real part over the range of frequencies (0.01 Hz–106 Hz). The resulting impedance spectra were analyzed with Z-view software (v2.8b, Scribner Associates Inc., Connecticut, NC, USA).

### 2.4. Descriptor Analysis

The 3D structures of the employed cations were generated via ChemAxon standardizer [36] from the SMILES notation available for each compound, and stored in SDF format. From these files, descriptors were generated using Mordred [37] and then processed using Weka 3.8.5 in order to select the most relevant descriptors based on their correlation with the photovoltaic or resistance parameters.

## 3. Results and Discussion

### 3.1. Preparation of Iodide Salts

Figure 2 illustrates the chemical structures of all the iodide-based organic salts employed in this study, which can be roughly divided into three categories: Alkylimidazolium (blue), picolinium (green) and tetra-alkylammonium and guanidinium (yellow). The imidazolium salts were prepared as described by Roma-Rodrigues et al. [38], through the simple alkylation with the corresponding alkyl iodide, to directly afford the iodide salt. The picolinium iodides were obtained through a similar method, albeit with the use of microwave heating, as described in the Appendix A. Guanidinium salt **9** was prepared as described by Huang et al. [39], while **10** was obtained through the ionic exchange of commercially available Aliquat^®^ 336 with NaI [40]. Tetrabutylammonium iodide (TBAI, **11**) is commercially available.

### 3.2. Photovoltaic Performance

The various iodide-based organic salts were employed as additives in prototype devices, by adding 0.5 M of the respective IL to an electrolyte composition of 0.5 M LiI and 0.05 M I_2_ in a mixture of ACN/VCN (85:15% *v*/*v*). A reference device was used (in non-optimized conditions) for comparison purposes, which did not contain ILs in its composition, using 1 M LiI instead. In all devices, conventional ruthenium dye N719 was employed as photosensitizer. The obtained current density-voltage (***J***-***V***) curve and its respective photovoltaic parameters are presented in Table 2 and Figure 3, respectively.

Overall, the addition of the ILs in the electrolyte composition did not result in a substantial change in the overall efficiency when comparing to the reference. The most notable exceptions to this trend are **E3** and **E6**, which show a more significant improvement. For most of the employed salts, the specific shift of the photovoltaic parameters is observed to be a slight gain in open circuit voltage (***V_oc_***) as well as fill factor (***FF***), which generally results from an increase in resistance parameters associated with electron recombination.

A trend of reduction in ***V_oc_***, ***J_sc_*** and ***FF*** with increase in chain length can be seen for both the MIM and DMIM cation structures, albeit to a lesser extent in the latter case while the opposite trend regarding ***V_oc_*** can be observed for the 3-picolinium family. This latter tendency might be a consequence of the 3-substituted pyridiniums’ hindered lateral assembly on the TiO_2_ surface, which in turn means that the alkyl chain size has a more pronounced effect on the spatial arrangement of the IL cations, consequently, the ***V_oc_***. Inversely, the symmetrical 4-substituted pyridinium equivalents can more efficiently assemble laterally, with the charged polar head of the pyridinium lying closer to the photoanode surface. This is in agreement with previously reported results from Jeon et al. [41], where they found that extending the length of the alkyl chain of alkylpyridinium iodides did not result in a significant change in ***V_oc_***. Meanwhile, the shift in ***J_sc_*** follows the same trend as the first family, showing a decrease in both parameters with the increase in chain length. This trade-off is consistent with a more efficient blocking of the TiO_2_ layer by the bulkier picoliniums, which will more effectively restrict the adsorption of the Li^+^ ions. The adsorption of small cations, such as Li^+^, on the TiO_2_ surface possesses a well-known effect on the energy level of its conduction band edge [42]. The shift to more positive energy levels leads to an increase of the injection driving force, which directly translates in a gain in photocurrent.

**E10** and **E11** showed a very different behavior from all other employed salts. Despite showing two of the highest values for ***V_oc_***, they also presented the lowest ***FF*** values by a wide margin. Additionally, they possess the lowest and highest photocurrent values of the entire set, respectively, which results in the relative worst efficiency for **E10**, and a very slight gain over the reference for **E11**. This stark difference from the other groups might result from the different type of steric hindrance possessed by these salts, which unlike the linear alkyl chains of the imidazoliums or picoliniums, possess a wider and bulkier steric effect.

Given the promising photovoltage, and photocurrent in the case of **E11**, the inclusion of the additive 4-TBP in the electrolyte mixture was considered as a potential solution for the underwhelming ***FF*** values. When included in the electrolyte composition, a significant gain in photovoltage and ***FF*** can be observed, which translates to a noticeably higher efficiency. This effect, as extensively reported in the literature [43,44,45], can be ascribed to 4-TBP adsorption on the TiO_2_ surface, causing a shift of the conduction band of TiO_2_ to more negative values, as well as an effective suppression of recombination from the injected electrons with the redox mediator. Consequently, this will also result in a reduction of the injection driving force, which will be negatively reflected in the photocurrent value.

As such, a second set of test devices, employing a similar composition of 0.5 M LiI, 0.5 M IL and 0.05 M I_2_ in ACN/VCN (85:15% *v*/*v*), with an additional 0.5 M 4-TBP, was prepared and analyzed. A cell employing 1M LiI (with no IL addition), but with the same 0.05 M I_2_ in ACN/VCN (85:15% *v*/*v*) and additional 0.5 M 4-TBP, was used as reference. The ***J***-***V*** curves and the resulting photovoltaic parameters are presented in Table 3 and Figure 4, respectively.

Inclusion of 4-TBP within the electrolyte causes, as anticipated, a significant increase in photovoltage parameters and ***FF***, as well as an expected trade-off in photocurrent, leading to an average efficiency increase of 1.7%. Compared to the reference, all employed electrolytes either matched or improved upon the gain in photovoltage, with the sole exception of **E8**. Additionally, most of them did not result in a further decrease of ***J_sc_*** (except **E5**, **E7** and **E10**).

The imidazolium family showed the most significant enhancement in the presence of 4-TBP, showing an average improvement of 213 mV in ***V_oc_*** and 0.08 in ***FF***, which translates to an average gain of efficiency of 2.13%. The significant improvement over the exclusive use of 4-TBP is reminiscent of the synergistic effect often observed when combining it with another additive, guanidinium thiocyanate (GuNCS) [46,47]. The guanidinium cations reportedly adsorb onto TiO_2_ and shift the conduction band to more positive values as a consequence of the accumulation of positive charge [48], similarly to Li cations. However, this additive also possesses the property of passivating the semiconductor surface and preventing recombination, which mitigates the photovoltage loss associated with cation adsorption [49]. When combined, these two components can lead to gains in both photovoltage and photocurrent, resulting in overall higher efficiencies.

The picolinium scaffolds constitute an interesting case, where the ethyl substituted salts (**E5** and **E7**), despite showing a noticeable increase in ***V_oc_*** of roughly 219 mV, suffer the most pronounced loss of ***J_sc_***, resulting in a comparatively lessened gain of efficiency. Curiously, the hexyl picoliniums show somewhat of an inverse behavior, showing the lowest gain of ***V_oc_*** and some of the lowest loss of ***J_sc_***. The increased bulk of these cations might lead to competition with 4-TBP for the available adsorption sites on the TiO_2_ surface [41], hindering the additives’ effect, while allowing the much smaller Li^+^ to pass through and adsorb.

As intended, **E11** improved significantly regarding its ***FF*** value upon addition of 4-TBP, showing the most pronounced increase of 0.11. Despite this, the observed gain of ***V_oc_***, and consequently efficiency, was underwhelming when compared with the remaining iodides. In contrast, **E10** showed no improvement of ***FF*** while also suffering a severe loss in photocurrent parameters, resulting in the lowest efficiency increase of 0.55%, less than a third of the average of the tested electrolytes. This unusual interaction warranted further study, which prompted us to analyze the test devices with electrochemical impedance spectroscopy, in order to investigate their interfacial processes [50,51].

### 3.3. Electrochemical Impedance Measurements

EIS allows the analysis of the interfacial photoelectrochemistry at the various regions of the photovoltaic device, clarifying the effects of the electrolyte components on these processes. The measurements were performed in dark conditions under a forward bias depending on the open circuit potential (OCP) of the cell. In the obtained Nyquist plots (Appendix A), two semicircles can be observed: a small semicircle within the high-frequency region, which can be attributed to the charge transfer between the Pt counter-electrode and the electrolyte (***R_Pt_***), and a larger circle at the middle frequency region, which represents the electron transfer at the TiO_2_ surface, including the electron diffusion in the semiconductor and the electron back reaction to the oxidized redox species (***R_ct_***). Consequently, ***R_ct_*** can be interpreted as the recombination resistance. The equivalent circuit model used for fitting the data is shown in Figure 5 and is composed of a series resistance (***R_s_***), the charge transfer resistances at the Pt (***R_Pt_***) and TiO_2_ (***R_ct_***) anodes, as well as constant phase elements (**CPE1** and **CPE2**), representing the interfacial capacitance.

The electron lifetime can also be obtained from the Bode phase plots (Appendix A), where two distinguishing frequency peaks can be detected. The peak at roughly 10 Hz can be attributed to the charge transfer processes present in the TiO_2_/electrolyte interface, while the peak in the 10^3^–10^4^ Hz region is indicative of the charge transfer at the Pt counter-electrode [52,53,54]. From the peak at 10 Hz, the frequency corresponding to the maximum angular frequency (ωm), the electron lifetimes (***τ_e_***) can be obtained according to Equation (1) [55]:(1)$$\tau =\frac{1}{f_{p}}$$

Table 4 shows the impedance parameters obtained for both sets of electrolytes with and without 0.5 M of 4-TBP, respectively.

By comparing the obtained results, the superiority of the imidazoliums in comparison to the remaining electrolytes becomes more apparent. They present, on average, higher values of ***R_ct_*** and ***τ_e_***, indicative of suppressed electron recombination between the injected electrons and the electrolyte, which will reflect positively on the ***V_oc_*** value and, consequently, the conversion efficiency. Additionally, they generally possess the lowest ***R_Pt_*** values, a necessary property to ensure effective reduction of the redox-shuttle present in the electrolyte, and in turn regeneration of the photo-oxidized dye [56]. **E10** and **E11** readily exemplify this, as their high ***R_Pt_*** values appear to be a likely cause for their subpar ***FF*** values. Interestingly, both electrolytes show near identical **FF**, despite the former having roughly twice the ***R_Pt_*** of the latter, suggesting an upper limit of resistance above which the ***FF*** parameter is no longer diminished. A trend of increasing ***R_ct_***, ***R_Pt_*** and ***τ_e_*** with increasing chain length can be observed for both the imidazolium and picolinium cations, consistent with a more effective passivation of the electrode surfaces by the bulkier cations.

The inclusion of 4-TBP has the readily apparent effect of increasing both the platinum counter-electrode resistance, the TiO_2_ diffusion/recombination resistance and the electron lifetime, a result not only of the previously mentioned adsorption/passivation of the semiconductor surface, but also of the platinum cathode [57]. The only exceptions to this increase are **E11**, which experiences a decrease across all parameters, and **E6,** which displays a decrease of ***R_ct_***. Despite the picoliniums showing a greater overall increase in ***R_ct_*** and ***τ_e_***, the imidazoliums are the ones that demonstrate the best balance between the beneficial increase in these two parameters, versus the deleterious increase in ***R_Pt_***, consequently achieving a higher gain in efficiency.

The case of **E11** is particularly interesting, given how ***R_ct_*** and ***τ_e_*** are halved in the presence of 4-TBP. Once again, this might result from competition between both additives for the photoanode surface, as reported by Zhang et al. [58], leading to an decrease of the presence of the bulkier TBA cation on this interface and, consequently, lower ***R_ct_*** and ***τ_e_*** [59].

### 3.4. Descriptor Analysis

To further rationalize the obtained results, several descriptors were calculated with Mordred [37] based on the 3D structure of the employed cations. Using Weka 3.8.5, descriptors were selected based on their correlation with photovoltaic and resistance parameters, which resulted in the selection of three main descriptors: Relative positive charge (***RPCG***) [60,61], molecular polarizability (***apol***) [62] and partition coefficient (***logP***) [63]. The clear connection between all the descriptors and the polarity of the cation structure underlines the importance of this property on the electrolyte’s performance.

The descriptors showed moderate to good exponential correlations with the counter-electrode resistance (***R_Pt_***) in the absence of 4-TBP, as can be seen in Figure 6. The inverse relationship between counter electrode resistance and the cation charge density is consistent with findings from Wang et al. [64], where the use of varying inorganic iodides resulted in a similar trend. Given the previously mentioned importance of this resistance parameter, the ability to roughly estimate its magnitude from a molecular descriptor is a valuable tool in the design of future electrolyte mixtures.

It was also possible to observe a general trend of decreasing ***V_oc_*** with increasing molecular polarizability in the 4-TBP set (Figure 7). This decrease curiously plateaus at roughly 650 mV (excluding **E8**), which corresponds to the photovoltage value of the reference mixture. This once again might be suggestive of a synergistic effect between the salts and 4-TBP, particularly for cations with low polarizability, which is in agreement with the known synergy between 4-TBP and GuNCS [46], another small cation with low polarizability.

The use of organic salts and ILs as electrolytes or redox mediators for application in DSSCs [65,66] is an interesting topic for academia as well as industry and further investigation is mainly required.

## 4. Conclusions

Iodide-based organic salts of various classes, including methyl and dimethylimidazolium, 3- and 4-picolinium, hexa-alkylguanidinium and tetra-alkylamonium, were prepared and employed as alternative iodide sources in electrolyte mixtures for DSSC devices. In general, a reduction in ***V_oc_***, ***J_sc_*** and ***FF*** was observed with the increase of the chain length of imidazolium-based electrolytes. Regarding ***V_oc_***, the opposite trend was observed for the 3-picolinium electrolytes, while in the case of 4-picolinium essentially no change was present. The addition of 4-TBP to the electrolyte formulation is beneficial in order to increase the photovoltage and ***FF*** parameters. The reason for this improvement can be related to a synergistic effect between this addtive and the imidazolium species. Inversely, the picolinium salts showed two clearly distinct behaviours depending on the presence of an ethyl or hexyl chain with the latter showing noticeably lower increases in ***V_oc_*** and decreases in ***J_sc_***, suggesting competitive behavior between this group of bulkier cations and 4-TBP. Electrochemical impedance (EIS) of the prepared devices allowed us to highlight the key role of the counter electrode resistance (***R_Pt_***) on device performance, namely its inverse relationship with the ***FF*** parameter. It is important to emphasize that the correlation between this parameter and the molecular structure of the employed cations was then clarified through the use of various molecular descriptors, namely ***RPCG***, ***apol*** and ***LogP***, where it was found that ***R_Pt_*** decreases with increasing charge density. Further exploration of chemoinformatic and machine-learning methods for the purpose of structure-performance clarification of electrolyte formulations is a promising research avenue for the further increase of DSSC efficiency.

The discovery of stable and sustainable electrolyte formulations is crucial for higher performances in DSSCs and in this context it is clear that organic salts and ILs can be useful as efficient additives.

## Figures and Tables

**Figure 1 nanomaterials-12-02988-f001:**
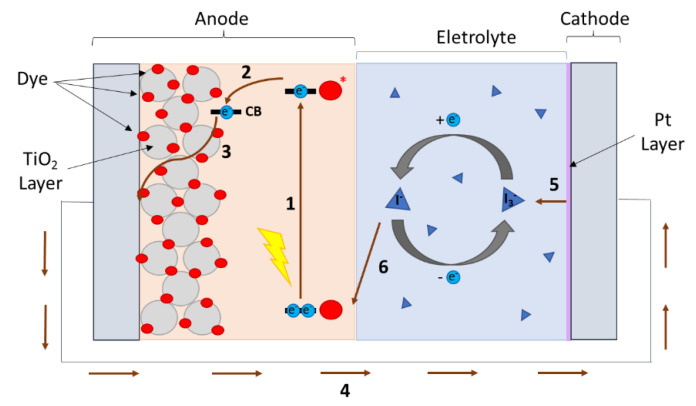
Schematic representation of the operating principles of a DSSC device.

**Figure 2 nanomaterials-12-02988-f002:**
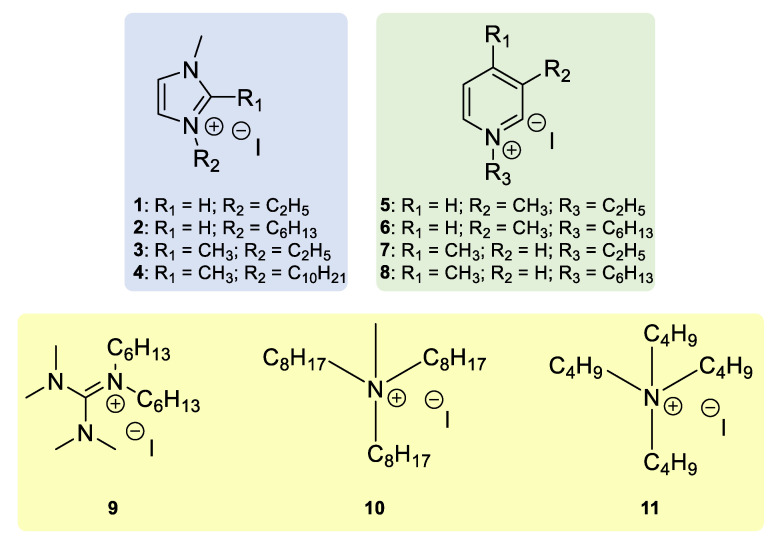
Chemical structures of the iodide-based organic salts synthesized and employed as additives in this work.

**Figure 3 nanomaterials-12-02988-f003:**
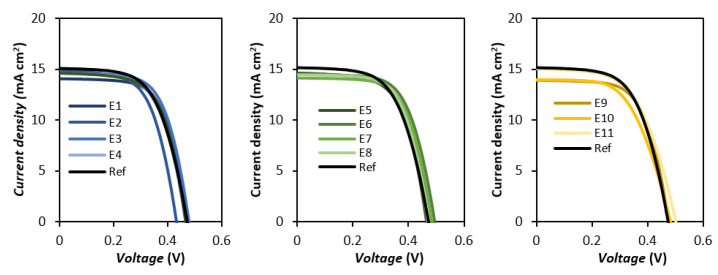
***J***-***V*** curves of the test cells employing the various the iodide-based organic salts as additives under 100 mW cm^−2^ simulated AM 1.5G illumination. The above results correspond to the best performing cell.

**Figure 4 nanomaterials-12-02988-f004:**
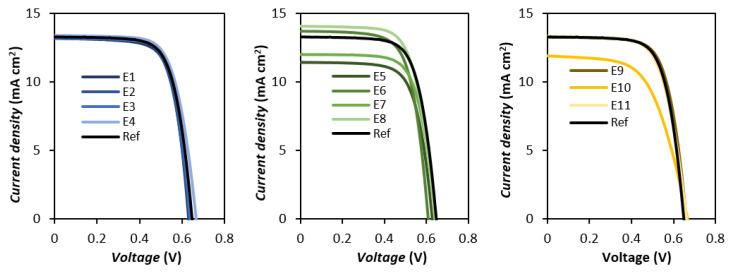
***J***-***V*** curves of the test cells employing the various iodide-based organic salts and 0.5 M 4-TBP as additives, under 100 mW cm^−2^ simulated AM 1.5G illumination. The above results correspond to the best performing cell.

**Figure 5 nanomaterials-12-02988-f005:**
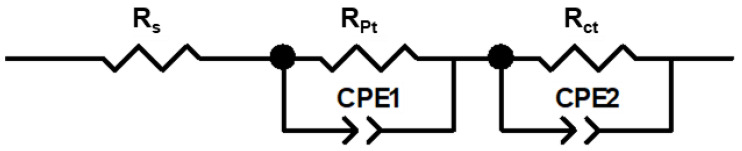
Representation of the equivalent circuit used to fit the EIS data.

**Figure 6 nanomaterials-12-02988-f006:**
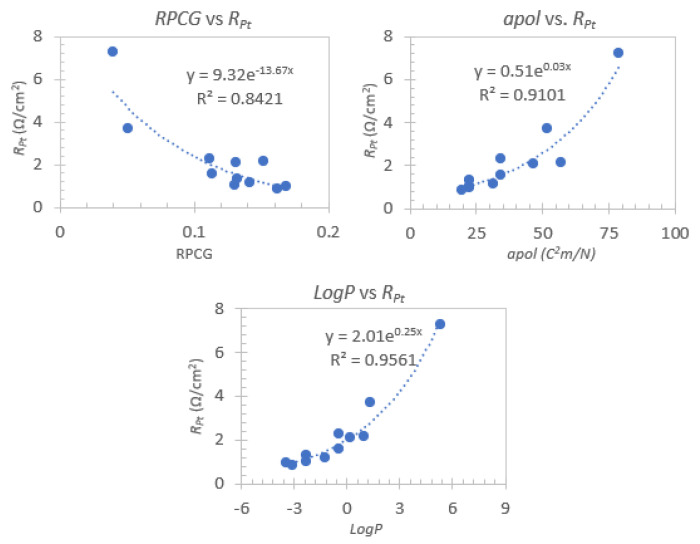
Exponential correlations between the counter-electrode resistance (***R_Pt_***) and the relative positive charge (***RPCG***), molecular polarizability (***apol***) and partition coefficient (***logP***) descriptors for the 4-TBP free set.

**Figure 7 nanomaterials-12-02988-f007:**
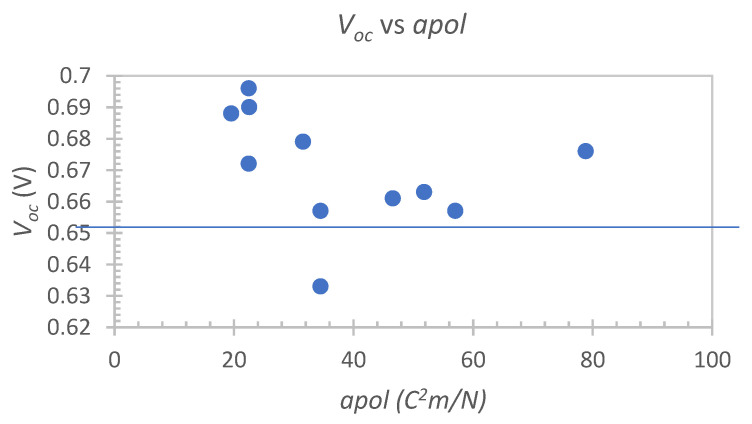
Exponential correlations between the counter-electrode resistance (***R_Pt_***) and the relative positive charge (***RPCG***), molecular polarizability (***apol***) and partition coefficient (***logP***) descriptors for the 4-TBP free set.

**Table 1 nanomaterials-12-02988-t001:** Photovoltaic parameters of several examples of IL electrolytes as pure solvents (top) and as additives in other organic solvents (bottom).

Electrolyte Mixture	Dye	*V_oc_* (V)	*J_sc_* (mA/cm^2^)	*FF*	*η* (%)	Ref.
**Solvent**						
I_2_/NBB/GuNCS in [DMIM]I/[C3MIM]I/[EMIM][TCB]	Z907Na	0.741	14.26	0.77	8.20	[22]
I_2_/NBB in [C3MIM]I/EMImTCM	Z907Na	0.752	12.81	0.76	7.40	[23]
I_2_/NMBI/LiI/GuNCS in HeMImI	N3	0.725	13.52	0.70	6.82	[24]
**Additive**						
[DMIM]I/I_2_/GuNCS/4-TBP/LiI in ACN/VN (85/15)	C101	0.778	17.94	0.79	11.0	[26]
[DMIM]I/I_2_/NBB/GuNCS/NaI in BN	C106	0.733	17.90	0.76	10.0	[27]
[C3MIM]I/[MIm-TEMPO][TFSI]/NOBF_4_/LiTFSI/NBB in MPN	D205	0.729	18.4	0.61	8.2	[28]

Note: NBB = N-Butylbenzimidazole; GuNCS = guanidinium thiocyanate; EMImTCM = 1-ethyl-3-methylimidazolium tricyanomethanide; NMBI = N-methylbenzimidazole; HeMImI = 1-(3-hexenyl)-3-methyl imidazoliums; BMTrI = 1-methyl-3-butyl-1,2,3-triazolium iodide; BN = butyronitrile [MIm-TEMPO] [TFSI] = 1-methyl-3-(2-oxo-2-(2,2,6,6-tetramethyl-1-oxyl-4-piperidoxyl) butyl)imidazolium bis(trifluoromethanesulfonyl)imide.

**Table 2 nanomaterials-12-02988-t002:** Performance values of the test cells employing the various iodide-based organic salts as additives under 100 mW cm^−2^ AM 1.5G illumination. The results correspond to the average of five measurements of two cells per electrolyte mixture.

Name	Cation	*V_oc_* (V)	*J_sc_* (mA/cm^2^)	*FF*	*η* (%)
**E1**	[C2MIM]	0.465 ± 0.008	14.35 ± 0.24	0.64 ± 0.013	4.27 ± 0.04
**E2**	[C6MIM]	0.454 ± 0.022	14.00 ± 0.89	0.61 ± 0.010	3.85 ± 0.05
**E3**	[C2DMIM]	0.483 ± 0.008	14.78 ± 0.13	0.63 ± 0.011	4.48 ± 0.03
**E4**	[C10DMIM]	0.464 ± 0.005	14.57 ± 0.07	0.62 ± 0.007	4.20 ± 0.02
**E5**	[C2-3PIC]	0.461 ± 0.007	14.52 ± 0.09	0.64 ± 0.010	4.25 ± 0.03
**E6**	[C6-3PIC]	0.493 ± 0.011	14.13 ± 0.34	0.64 ± 0.009	4.45 ± 0.18
**E7**	[C2-4PIC]	0.470 ± 0.014	13.92 ± 0.27	0.65 ± 0.006	4.27 ± 0.19
**E8**	[C6-4PIC]	0.475 ± 0.006	13.89 ± 0.64	0.63 ± 0.020	4.16 ± 0.28
**E9**	[C6-TMG]	0.472 ± 0.007	13.81 ± 0.08	0.63 ± 0.006	4.13 ± 0.09
**E10**	[ALIQUAT]	0.492 ± 0.013	13.75 ± 0.21	0.56 ± 0.022	3.78 ± 0.11
**E11**	[TBA]	0.493 ± 0.008	15.08 ± 0.03	0.57 ± 0.004	4.21 ± 0.10
**Ref**	---	0.470 ± 0.005	14.55 ± 0.55	0.61 ± 0.021	4.15 ± 0.12

**Table 3 nanomaterials-12-02988-t003:** Performance values of the test cells employing the various iodide-based organic salts as additives in combination with 4-TBP under 100 mW cm^−2^ AM 1.5G illumination. The results correspond to the average of five measurements of two cells per electrolyte mixture.

Name	Cation	*V_oc_* (mV)	*J_sc_* (mA/cm^2^)	*FF*	*η* (%)
**E1**	[C2MIM]	0.688 ± 0.013	13.06 ± 0.15	0.71 ± 0.005	6.37 ± 0.06
**E2**	[C6MIM]	0.679 ± 0.010	13.41 ± 0.10	0.70 ± 0.007	6.39 ± 0.05
**E3**	[C2DMIM]	0.690 ± 0.008	13.05 ± 0.13	0.72 ± 0.005	6.48 ± 0.09
**E4**	[C10DMIM]	0.661 ± 0.008	13.26 ± 0.20	0.69 ± 0.009	6.08 ± 0.14
**E5**	[C2-3PIC]	0.672 ± 0.004	11.19 ± 0.25	0.70 ± 0.006	5.28 ± 0.14
**E6**	[C6-3PIC]	0.657 ± 0.009	13.36 ± 0.31	0.69 ± 0.012	6.07 ± 0.14
**E7**	[C2-4PIC]	0.696 ± 0.007	12.02 ± 0.04	0.69 ± 0.018	5.75 ± 0.10
**E8**	[C6-4PIC]	0.633 ± 0.007	13.94 ± 0.11	0.69 ± 0.010	6.13 ± 0.11
**E9**	[C6-TMG]	0.657 ± 0.009	13.40 ± 0.07	0.69 ± 0.018	6.09 ± 0.10
**E10**	[ALIQUAT]	0.676 ± 0.013	11.71 ± 0.26	0.55 ± 0.033	4.33 ± 0.27
**E11**	[TBA]	0.663 ± 0.013	13.08 ± 0.22	0.68 ± 0.016	5.90 ± 0.09
**Ref**	---	0.655 ± 0.008	13.07 ± 0.18	0.70 ± 0.005	6.00 ± 0.05

**Table 4 nanomaterials-12-02988-t004:** EIS parameters for the test cells employing the various iodide-based organic salts as additives, with (*) and without 4-TBP, obtained from the Nyquist plots.

Name	Cation	*R_s_* (Ω/cm^2^)	*R_Pt_* (Ω/cm^2^)	*R_ct_* (Ω/cm^2^)	*τ_e_* (ms)
**E1/E1 ***	[C2MIM]	1.363/1.140	0.855/1.486	8.581/12.046	4.57/7.88
**E2/E2 ***	[C6MIM]	1.380/1.224	1.152/1.740	12.154/16.158	6.03/9.54
**E3/E3 ***	[C2DMIM]	1.312/1.459	0.975/1.137	11.323/15.531	7.14/9.70
**E4/E4 ***	[C10DMIM]	1.014/1.594	2.080/2.495	16.523/17.273	10.27/10.87
**E5/E5 ***	[C2-3PIC]	1.636/1.308	1.318/2.483	6.860/16.654	3.05/8.36
**E6/E6 ***	[C6-3PIC]	1.479/1.135	1.554/2.258	10.684/9.829	5.24/5.90
**E7/E7 ***	[C2-4PIC]	0.965/1.675	1.019/2.768	7.462/9.624	3.72/5.55
**E8/E8 ***	[C6-4PIC]	1.611/1.511	2.297/3.269	10.016/13.508	5.30/7.99
**E9/E9 ***	[C6-TMG]	1.085/1.398	2.146/2.436	8.291/17.885	5.46/10.86
**E10/E10 ***	[ALIQUAT]	1.443/1.457	7.244/12.123	8.942/15.415	7.77/11.70
**E11/E11 ***	[TBA]	1.030/1.373	3.702/3.489	24.912/17.775	21.79/11.24
**Ref/Ref ***	---	1.483/1.434	1.076/1.501	7.338/7.56	6.09/5.54

## Data Availability

The data provided in this study are available in the article and Appendix A file submitted.

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
