# Peer review of "Effect of Iodide-Based Organic Salts and Ionic Liquid Additives in Dye-Sensitized Solar Cell Performance"

_nanomaterials, 2022, doi:10.3390/nano12172988_

Round 1
Reviewer 1 Report
The writing and presentation of the text are excellent.
However
Is the accuracy of the numbers in the table justified? Are the measurement tools accurate enough to pick those up?
The line of regression shown in Figure 6 can be easily rounded up there is no need to have a pre-factor and power law index of 2.0132 please consider rounding it up to 2.01
Have a phrase on how artificial intelligence may aid with formulations added to the text.
Author Response
The Reviewer 1 (R1) considered that “The writing and presentation of the text are excellent.”
Authors: Thank you for your comments and suggestions.
The following comments/suggestions were included as required:
R1: Is the accuracy of the numbers in the table justified? Are the measurement tools accurate enough to pick those up?
Authors: The values of the tables are presented with the required accuracy. It is important to note that all results correspond to the average of five measurements of two cells per electrolyte mixture. The correspondent errors were also included.
R1: The line of regression shown in Figure 6 can be easily rounded up there is no need to have a pre-factor and power law index of 2.0132 please consider rounding it up to 2.01
Authors: Pre-factors and power laws have been rounded as mentioned
R1: Have a phrase on how artificial intelligence may aid with formulations added to the text.
Authors: An additional sentence was included in Line 462 about the use of cheminformatics methods to help in the formulation optimization.
Reviewer 2 Report
Authors studied the effect of organic salts and ionic liquid additives on the performances of DSSCs. After reading Authors’ article, I suggest the below change and revision for improving this manuscript. 1) English style should be improved, 2) Manuscript format/sequence should be revised, and finally 3) Conclusion part should be revised clearly. A major revision is needed.
- Abstract is an independent part. Hence, unnecessary abbreviation should be deleted, e.g., DSSCs, RPCG, apol, LogP, etc.
- According to the author guideline, “The abstract should be a total of about 200 words maximum.” Please revise Abstract part briefly.
- Line 34: DSSC => DSSCs
- Check Reference position all the places of your manuscript, e.g., ,[3] => [3],
- Check the sequence of your writing, e.g., Materials and Methods should be before Results.
- Conclusions part: Authors 'once again' cited many references here (e.g., [66,65,37,39-41, and 61], but usually, in the Conclusions part, any citation is not necessary. Authors can point out what Authors found out in the study, briefly.
- Line 87. Check English
- Table 1. Physical parameters should be italic, e.g., Voc, Jsc, η, etc. (Not only Table 1, but also all the other places also)
- Line 105: 4-tertbutylpyridine addition => 4-tertbutylpyridine (4-TBP) addition.
o (Check others, when abbreviation is needed, Authors should introduce it when the word appears first).
o Line 69: ionic liquids => ionic liquids (ILs)
- Line 116: Results and discussion => Results and Discussion
- Line 130: Structure of => Chemical structure of
- Line 137: current-voltage (I-V) curve => current density-voltage (J-V) curve
- Table 2. Voc (mV) => Voc (V) (Note: for consistency to Figure 3, the voltage unit should be revised and accordingly, each values should be corrected in Table 2)
- Table 2 and Figure 3: Jsc’s unit should be consistent, mA/cm2 or mA cm^-2 (please use only one style)
- Line 158: the packing of the cation => (Describe it in a better way because it is in liquid state) Maybe, the packing of the cation of ionic liquid
- Line 159: chain facing away from the photoanode surface.=> (Explain the reasons more scientifically. Why away from the photoanode surface?)
- Line 179: 4-tert-butylpyridine (4-TBP) => 4-TBP
- What is a benefit of mixed solvent (e.g., acetonitrile/valeronitrile) instead of acetonitrile alone? Explain it in your experimental part.
- Figure 4 should be improved. 1) X-axis scale should be in detail. 2) The middle figure’s y-axis’s title is now partially overlapped with scale, 10.
- GuNCS => guanidinium thiocyanate (GuNCS) (for reader-friendly)
- Line 244: RPt => definition should be introduced in this spot.
- Figures 6 and 7. In the x-axis’ title, if there is unit or dimension, please include it. For example, polarizability has a SI-unit. Right?
- Line 327 to 334: Make it one paragraph.
- Line 335: DSSCs fabrication = > DSSC fabrication
- Line 403-428: Conclusion part is not acceptable.
Author Response
The Reviewer 2 (R2) considered that “Authors studied the effect of organic salts and ionic liquid additives on the performances of DSSCs. After reading Authors’ article, I suggest the below change and revision for improving this manuscript. 1) English style should be improved; 2) Manuscript format/sequence should be revised, and finally 3) Conclusion part should be revised clearly. A major revision is needed.”
Authors: Thank you for your comments and suggestions. The English style was improved in the manuscript and the conclusion part revised as suggested.
R2: Abstract is an independent part. Hence, unnecessary abbreviation should be deleted, e.g., DSSCs, RPCG, apol, LogP, etc.
Authors: Unnecessary abbreviations were removed from the abstract. Abbreviation of the ionic liquid was maintained ([C2DMIM]I) for brevity
R2: According to the author guideline, “The abstract should be a total of about 200 words maximum.” Please revise Abstract part briefly.
Authors: The abstract has been revised in order to be until 200 words.
R2: Line 34: DSSC => DSSCs
Authors: The Correction was made.
R2: Check Reference position all the places of your manuscript, e.g., ,[3] => [3],
Authors: All references were reformatted accordingly.
R2: Check the sequence of your writing, e.g., Materials and Methods should be before Results.
Authors: Sequence of text has been changed accordingly.
R2: Conclusions part: Authors 'once again' cited many references here (e.g., [66,65,37,39-41, and 61], but usually, in the Conclusions part, any citation is not necessary. Authors can point out what Authors found out in the study, briefly.
Authors: The references were deleted from conclusion part.
R2: Line 87. Check English
Authors: The Text was changed to the following: “In a separate work by Gao at al[27] direct comparison between electrolyte mixtures containing ILs as mixed solvents or as additives was described,(…)”
R2: Table 1. Physical parameters should be italic, e.g., Voc, Jsc, η, etc. (Not only Table 1, but also all the other places also)
Authors: All physical parameters were changed to italic format.
R2: Line 105: 4-tertbutylpyridine addition => 4-tertbutylpyridine (4-TBP) addition. (Check others, when abbreviation is needed, Authors should introduce it when the word appears first).
R2: Line 69: ionic liquids => ionic liquids (ILs)
Authors: Several abbreviations have been corrected. The first instance of abbreviation for ionic liquids is present in Line 62, as such any following instance have been changed to “IL” or “ILs”
R2: Line 116: Results and discussion => Results and Discussion. Line 130: Structure of => Chemical structure of
Authors: The corrections were made
R2: Line 137: current-voltage (I-V) curve => current density-voltage (J-V) curve
Authors: All instances of current-volage and I-V were corrected accordingly
R2: Table 2. Voc (mV) => Voc (V) (Note: for consistency to Figure 3, the voltage unit should be revised and accordingly, each values should be corrected in Table 2)
Authors: All Voc values have been translated from mV to V
R2: Table 2 and Figure 3: Jsc’s unit should be consistent, mA/cm2 or mA cm^-2 (please use only one style)
Authors: The units for Jsc have been corrected to mA/cm2 across the document
R2: Line 158: the packing of the cation => (Describe it in a better way because it is in liquid state) Maybe, the packing of the cation of ionic liquid
Authors: Replaced “the packing of the cation” with “the spatial arrangement of the IL cations”
R2: Line 159: chain facing away from the photoanode surface.=> (Explain the reasons more scientifically. Why away from the photoanode surface?)
Authors: Replaced “Inversely, the symmetrical 4-substituted pyridinium equivalents can assemble laterally, with the alkyl chain facing away from the photoanode surface.” to “Inversely, the symmetrical 4-substituted pyridinium equivalents can more efficiently assemble laterally, with the charged polar head of the pyridinium lying closer to the photoanode surface.
R2: Line 179: 4-tert-butylpyridine (4-TBP) => 4-TBP
Authors: All mentions of 4-tertbutylpyridine after the initial abbreviation have been replaced with 4-TBP
R2: What is a benefit of mixed solvent (e.g., acetonitrile/valeronitrile) instead of acetonitrile alone? Explain it in your experimental part.
Authors: An explanation for the electrolyte formulation has been included in Line 163
R2: Figure 4 should be improved. 1) X-axis scale should be in detail. 2) The middle figure’s y-axis’s title is now partially overlapped with scale, 10.
Authors: X-axis have been changed to show increments of 0.2 V instead, as in the case of Figure 3. The Y-axis titles have also been adjusted more consistently
R2: GuNCS => guanidinium thiocyanate (GuNCS) (for reader-friendly)
Authors: The first instance of abbreviation can be found in the footnote of Table 1 (Line 110), but another instance has been added in Line 290 to simplify reading
R2: Line 244: RPt => definition should be introduced in this spot.
Authors: The Definition of RPt is within the sentence “a small semicircle within the high-frequency region, which can be attributed to the charge transfer between the Pt counter-electrode and the electrolyte (RPt)”
R2: Figures 6 and 7. In the x-axis’ title, if there is unit or dimension, please include it. For example, polarizability has a SI-unit. Right?
Authors: Polarizability units (C2m/N) were added. The remaining two descriptors do not possess units
R2: Line 327 to 334: Make it one paragraph. Line 335: DSSCs fabrication = > DSSC fabrication
Authors: The Corrections were made
R2: Line 403-428: Conclusion part is not acceptable.
Authors: The conclusion part was changed as suggested. The references were placed in the end of discussion part. The main conclusions of the work were considered as well as future perspectives in the last two sentences.
Round 2
Reviewer 2 Report
Based on Authors' revision according to Reviewer's comments and suggestions, I recommend this manuscript should be published as it is.